# Resource depletion through primate stone technology

**Lydia V Luncz[1]\*, Amanda Tan[2], Michael Haslam[3], Lars Kulik[4], Tomos Proffitt[3], Suchinda Malaivijitnond[5,6], Michael Gumert[2,3,5]**

[1]Institute for Cognitive and Evolutionary Anthropology, School of Anthropology and Museum Ethnography, University of Oxford, Oxford, United Kingdom; [2]Division of Psychology, School of Humanities and Social Sciences, Nanyang Technological University, Singapore, Singapore; [3]Primate Archaeology Research Group, School of Archaeology, University of Oxford, Oxford, United Kingdom; [4]Max Planck Institute for Evolutionary Anthropology, Leipzig, Germany; [5]Department of Biology, Faculty of Science, Chulalongkorn University, Bangkok, Thailand; [6]National Primate Research Center of Thailand-Chulalongkorn University, Saraburi, Thailand

**Abstract** Tool use has allowed humans to become one of the most successful species. However, tool-assisted foraging has also pushed many of our prey species to extinction or endangerment, a technology-driven process thought to be uniquely human. Here, we demonstrate that tool-assisted foraging on shellfish by long-tailed macaques (*Macaca fascicularis*) in Khao Sam Roi Yot National Park, Thailand, reduces prey size and prey abundance, with more pronounced effects where the macaque population size is larger. We compared availability, sizes and maturation stages of shellfish between two adjacent islands inhabited by different-sized macaque populations and demonstrate potential effects on the prey reproductive biology. We provide evidence that once technological macaques reach a large enough group size, they enter a feedback loop – driving shellfish prey size down with attendant changes in the tool sizes used by the monkeys. If this pattern continues, prey populations could be reduced to a point where tool-assisted foraging is no longer beneficial to the macaques, which in return may lessen or extinguish the remarkable foraging technology employed by these primates.

DOI: https://doi.org/10.7554/eLife.23647.001

\*For correspondence: Lydia. Luncz@anthro.ox.ac.uk

**Competing interests:** The authors declare that no competing interests exist.

## Introduction

Humans are currently contributing to one of the most dramatic extinction events in global history (*Barnosky et al., 2011*). In coastal areas, a large component of this pressure comes from our use of increasingly efficient technologies for harvesting food (*Jackson et al., 2001*) as well as significantly denser human populations (*Small and Nicholls, 2003*). For shellfish, however, archaeological evidence demonstrates that over-harvesting is not a new phenomenon, with intertidal species repeatedly depleted in various parts of the world (*Mannino and Thomas, 2002*; *Cortés-Sánchez et al., 2011*). Observations and examinations of past and present human populations that exploit shellfish have revealed that over-harvesting and focused collection on the larger individuals, shortens the life histories of the shellfish that are preyed on, resulting in significant reduction in shellfish sizes (*Blackburn et al., 2004*; *Erlandson et al., 2008*; *Fenberg and Roy, 2008*; *Langejans et al., 2012*; *Morrison and Hunt, 2007*; *Spennemann, 1987*; *Branch and Odendaal, 2003*; *Parkington, 2008*). Shellfish exploitation has also been linked to the cognitive, social and technological changes that led to the emergence of modern human behavior (*Marean et al., 2007*; *Will et al., 2016*).

Recently, island-dwelling wild Burmese long-tailed macaques (*Macaca fascicularis aurea*) were also found to regularly exploit shellfish with the aid of stone tools (*Malaivijitnond et al., 2007*). The

**eLife digest** Tools have helped us to become one of the most successful species on Earth. However, our use of tools for hunting and foraging has also caused many prey species to become endangered, or even extinct. In some cases, it has also led to evolutionary changes in prey species. For example, over-harvesting of shellfish in coastal areas has driven the shellfish to become smaller in size.

Recently, long-tailed macaques living on islands off the coast of Thailand and Myanmar were also found to use stone tools to forage on shellfish. The macaques use these tools to break open oysters, snails and other prey on the seashore. Studying these monkeys offers the opportunity to test how a non-human primate using stone-based technology affects the sustainability of their prey species.

Luncz et al. investigated how foraging with stone tools by long-tailed macaques living in Khao Sam Roi Yot National Park in Thailand affects local shellfish populations. This revealed that macaques using stone tools alter prey populations in a similar way to human technologies. Specifically, tool use by the macaques significantly reduced the numbers and size of the prey, especially on islands that were home to larger populations of monkeys. In return, the macaques responded by using smaller and smaller stone tools. This "feedback loop" could lead to the stone tools becoming less useful to the macaques to the point where they stop using them.

An important next step is to learn whether continued foraging of shellfish might actually lead to the macaques losing the knowledge on how to use stone tools. Luncz et al. propose that since stone tools first emerged, the size of the tools and the prey species they target may have been gradually decreasing. Future archaeological investigations will clarify if this is indeed the case.
DOI: https://doi.org/10.7554/eLife.23647.002

macaques use these tools to break open oysters, gastropods, and other intertidal prey (*Gumert and Malaivijitnond, 2012*), during intensive foraging episodes that can result in dozens of shellfish being eaten by a single animal using a single tool (*Haslam et al., 2016a2016*). Protein and nutrients obtained in this manner would otherwise be inaccessible to the macaques, providing significant benefits under closed island conditions. Stone tool behavior is currently known from islands off the west coast of Thailand and Myanmar, as well as two islands on the east coast of Thailand (*Tan and , 2017*; *Gumert and Malaivijitnond, 2013*), where Burmese long-tailed macaques have hybridized with common long-tailed macaques (*M. f. fascicularis*) (*Bunlungsup et al., 2016*). The study of island populations is particularly valuable for detecting the effects of predators on local prey species, because of the relatively closed nature of island ecological systems (*Blackburn et al., 2004*; *Swadling, 2010*). The macaques therefore offer the opportunity to test the effect of non-human stone technology on resource biology and sustainability.

Here, we studied a population of long-tailed macaques in Khao Sam Roi Yot National Park, on the east coast of Thailand, that are hybrids of common and Burmese long-tailed macaques. Two groups of macaques live on two neighboring islands – Koram and NomSao – which are separated by less than 400 m, and therefore the overall environmental conditions under which shellfish grow on each island are similar. Koram, however, is densely populated and has a total of at least 80 macaques, while NomSao is sparsely populated by only 9 individuals. On both islands, the macaques are provisioned with food and water, cared for during times of resource scarcity, and thus sustained by human intervention. Despite this, the macaques forage daily for shellfish along the shorelines, like their counterparts on the west coast of Thailand.

To investigate the potential effects of macaque stone technology on the sustainability of marine prey, we compared tool use behavior and shellfish properties between the islands, in addition to environmental variables that may affect shellfish size. Wild macaques have been reported to match their tool sizes to prey size (*Gumert and Malaivijitnond, 2013*), although this effect has not yet been reported for our study area. We therefore recorded the weight of tools used for respective prey species and compare tool selection patterns between the two foraging groups. To control for environmental factors potentially influencing tool selection, we further compared available shorelines, the local stone availability and stone sizes between both islands.

To determine whether macaque tool-assisted predation follows the same trajectory of resource depletion seen in anthropogenic contexts, we compared shellfish densities and sizes between the two islands. We concentrated on the prey most commonly targeted by the macaques: rock oysters (*Saccostrea cucullata*), tropical periwinkle (*Planaxis sulcatus*), bifasciated cerith (*Clypeomorus bifasciatus*), and tooth-lipped snail (*Monodonta labio*). This approach allowed us to assess whether macaques follow a size-selective harvesting of prey species. The harvesting of large individuals has been reported to leave prey populations with fewer individuals able to reproduce, which additionally hinders population stability and recovery (*Morrison and Allen, 2017*). Depending on the life cycle of prey species, size selective harvesting in wild macaques might therefore affect the likelihood of resource depletion. Additionally, species with high reproductive rates, especially at an early age, are reported to be relatively resistant to overharvesting. Prey with high localized aggregation are more vulnerable to foraging, as this clustering enables higher predator foraging efficiency per time (*Morrison and Allen, 2017*). We therefore used published life history data (*Angell, 1986*; *Rohde, 1981*; *Moore, 1937*; *Ohgaki, 1997*) to more accurately judge the impact of tool-assisted foraging might have on marine prey populations.

Over-harvesting marine prey has been reported to alter the life history of shellfish as an evolutionary response to increased selection pressure (*Fenberg and Roy, 2008*). We therefore compared sexual maturation stages of different shellfish sizes between the islands to assess potential changes in prey life history where predation pressure is high. As a final measure of possible environmental influence, we assessed the rate of resource depletion through tool-assisted foraging by recording the number of prey items consumed during daily foraging events. The estimated rate of depletion could then be judged against the calculated shellfish population on each island.

Based on analogy from the human record, we hypothesized that macaque stone technology will exert significant pressure on local shellfish sustainability on the island with the largest number of predators, Koram. We tested this hypothesis through the following predictions. If stone tool size selection by macaques at Khao Sam Roi Yot reflects that seen previously elsewhere, then we expect that stone tools on the two islands would match their respective shellfish prey size, with larger tools selected for larger prey. Additionally, if macaques follow patterns seen in human shellfish foraging (*Mannino and Thomas, 2002*), we expect that the macaques would preferentially target larger prey, which would lead to smaller prey available on the heavily populated Koram Island than on NomSao Island. If resource depletion occurred on these islands because of pressure from tool-using macaques, we further expect that shellfish on Koram would be less available than on NomSao. Finally, if the macaques have had an evolutionarily significant effect on prey biology, we would expect to find that shellfish on Koram, which are exposed to increased predation pressure, mature at a younger age (and smaller size) than shellfish on NomSao island.

If we find our predictions to be true, this may be an indication that macaque stone technology could lead to a depletion of prey populations, mirroring the effects of human predation seen in the archaeological and ethnographic records (*Mannino and Thomas, 2002*; *Cortés-Sánchez et al., 2011*).

## Results

During the observation period for this study, 26 long-tailed macaques on Koram Island and 4 macaques on NomSao Island regularly used stone tools to forage for shellfish. Since the NomSao group contained only adult males, we limited our comparative analyses with Koram Island to tools used by males only (Koram N = 14, NomSao N = 4). The tool behavior of macaques was different between the two islands. We found that the two macaques groups select different sized stone tools for shellfish foraging. Tools selected by macaques to open marine gastropods were significantly smaller on Koram Island than on NomSao Island (*Figure 1A*, *Figure 1—source data 1*; LM: N = 67, E = −0.585, SE = 0. 113, $F_{(1,67)}$= 26.645, p<0.001). We found the same result for the stones tools used to crack open oysters; tool used on Koram were smaller than on NomSao (*Figure 1B*, *Figure 1—source data 1*; N = 186, LMM: E = −1.729, SE = 0.562, $\chi^2$ = 6.040, df = 1, p=0.009). Permutation tests revealed the same p values for both models (LM (weight of snail tool) p<0.001; LMM (weight of oyster tool) p=0.009).

To investigate potential drivers for the observed difference in tool selection between the two islands, we compared environmental factors that may influence macaque tool selection. Shellfish

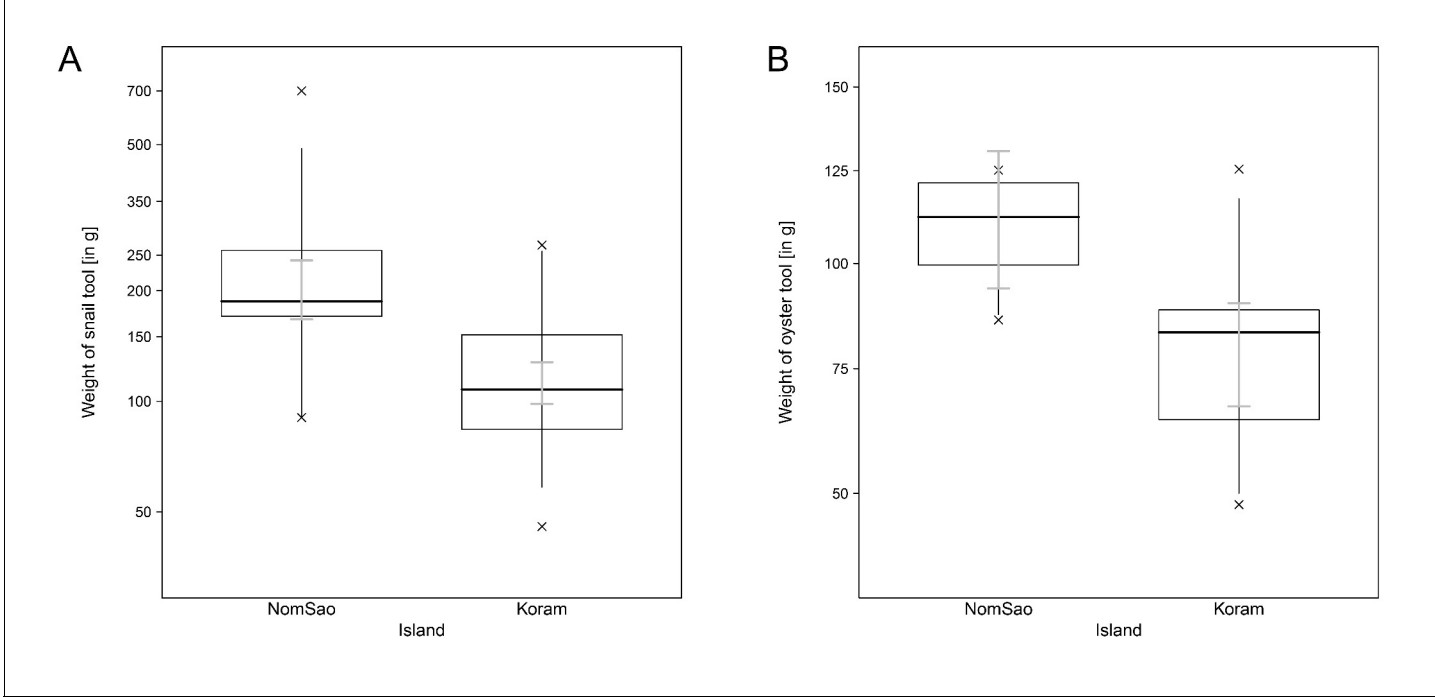

**Figure 1.** Tool weights of Koram and NomSao Islands. (**A**) Comparison between weights of stones used by macaques on Koram and NomSao Islands to open oysters. The plot shows all quantiles and the CIs (grey). (**B**) Comparison between weights of stones used by macaques on Koram and NomSao Islands to open snails. The plot shows all quantiles and the Cis (grey).

DOI: https://doi.org/10.7554/eLife.23647.003

The following source data is available for figure 1:

**Source data 1.** Stone tools used.
DOI: https://doi.org/10.7554/eLife.23647.004

foraging occurred mainly along the northwest coasts on both islands (*Figure 2*). The length of accessible shoreline suitable for shellfish foraging, however, differed between the two islands. On Koram Island, three independent rocky areas, separated by sandy patches, were suitable for shellfish foraging along a total shoreline distance of 1551 m. On average, this resulted in 55.4 m accessible shoreline per tool-using macaque (N = 26). On NomSao Island two rocky foraging zones covered a total of 653 m averaging 163.3 m suitable shoreline per tool user (N = 4). The length of available coastline for foraging is therefore almost three times larger on NomSao island per tool-using macaque.

The availability of stones suitable to use as tools did not differ between the islands (*Figure 3A*, *Figure 3—source data 1*; two sample t-test: N = 558, $t_{(19)}$ = 1.403, p=0.177). Stones available on NomSao, however, were significantly smaller than on Koram (*Figure 3B*, *Figure 3—source data 1*; two sample t-test: $t_{(440)}$ = −2.023, p=0.044). (Note that the oyster processing tools on NomSao were of a similar size to randomly available stones (*Figures 1B* and *3B*).

The availability of marine gastropods between the two islands differed significantly for two species, with a higher number of periwinkles and tooth-lipped snails on NomSao (*Figure 4*, *Figure 4—source data 1*; t-tests: *P. sulcatus*: N = 749 individuals, $t_{(18)}$ = 2.885, p=0.010, *M. labio*:: N = 72 individuals, $t_{(13)}$= 2.912, p=0.012). For bifasciated cerith, we found no difference between the two islands (*C. bifasciatus:* N = 72 individuals, $t_{(16)}$= −1.090, p=0.292). Oyster beds were abundant along the lengths of rocky shores of both islands and therefore were not considered a potential limiting foraging factor. However, we found that rock oysters were significantly larger on NomSao than on Koram island (*Figure 5A*, *Figure 5—source data 1*; two sample t-test: N = 1018 individuals, $t_{(563)}$ = 9.873, p<0.001). We found a similar result for two of the investigated marine gastropod species, the periwinkle and the cerith. Both prey species were significantly larger on NomSao (*Figure 5B*, *Figure 5—source data 2*; two sample t-tests: *P. sulcatus*: N = 223 individuals, $t_{(150)}$ = 19.929, p<0.001; *C. bifasciatus*: N = 218 individuals, $t_{(206)}$= 9.762, p<0.001). We did not find enough tooth-lipped

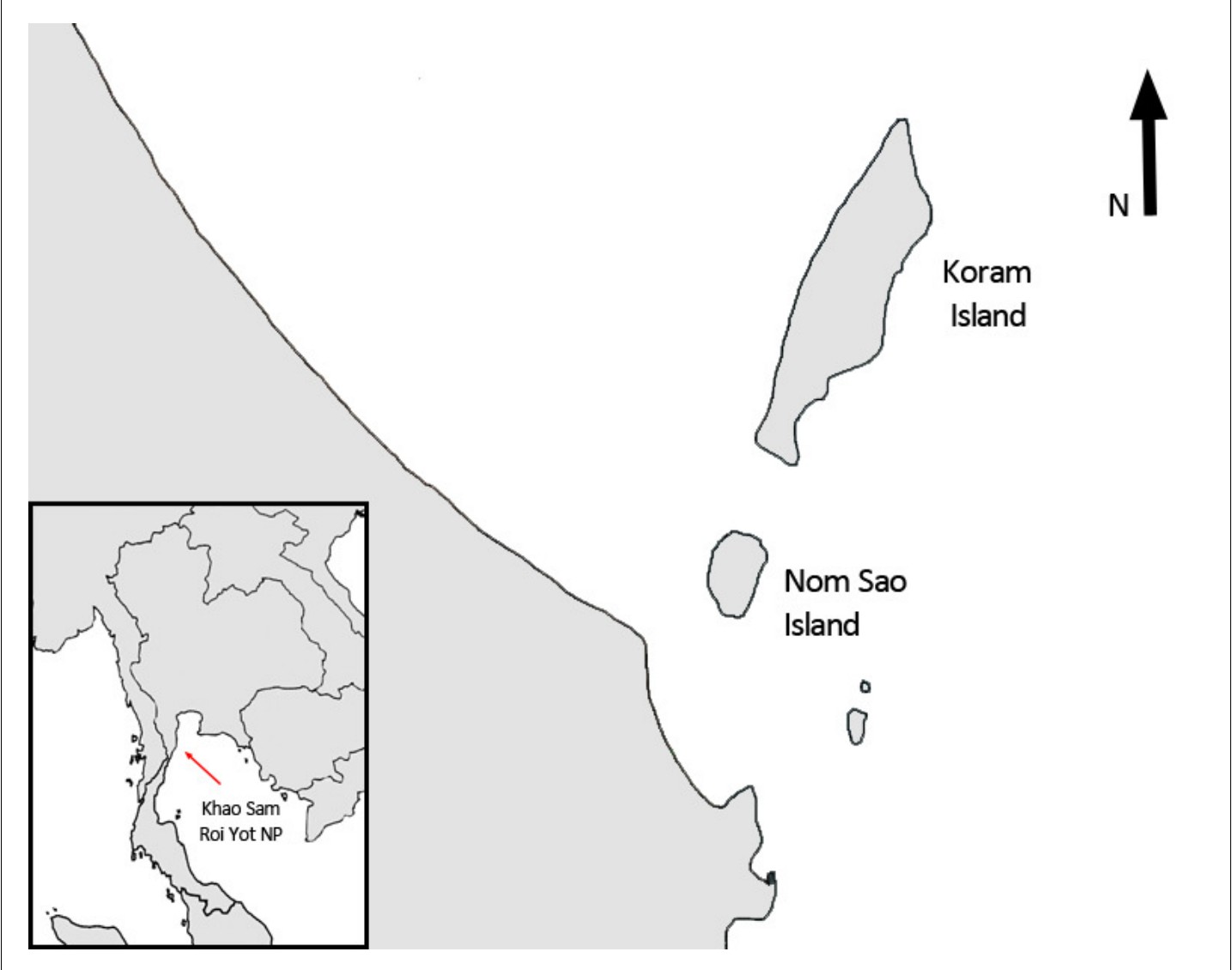

**Figure 2.** Location of the two study islands (Koram and NomSao) in Khao Sam Roi Yot National Park, Thailand.

DOI: https://doi.org/10.7554/eLife.23647.005

snails on Koram (N = 4, versus N = 68 on NomSao) to do a size comparison between the two islands (see *Figure 6B* for size differences).

Sizes of snails between both islands correlated with the maturation stage of these species. Across all collected snails on both islands, more mature specimens had larger shells (*Source data 1*, comparisons full with null model: LM: N = 77 individuals, $F_{(4,45)}$ = 61.130, p<0.001). Further, the snail size for a given maturation stage was not significantly different between Koram and NomSao Islands (LM: E = 0.319 SE = 0.261 $F_{(1,48)}$ = 1.494 P=0.228).

We assessed macaque predation pressure using observations of prey consumption on Koram Island. On average, one tool-using macaque on Koram Island consumes 46.5 shellfish items per day (36 rock oysters, 1.6 tropical periwinkle and 8.9 other species, *Source data 2*). In total, the studied monkey group on Koram Island (N = 26) consumes approximately 441,000 prey items per year, of which almost 61,000 are periwinkles (foraging items consumed per individual multiplied by the number of tool users per island multiplied by 4 hr foraging time per day for 365 days per year). Focusing on periwinkles, the four tool-using macaques on NomSao Island would in theory consume some 9344 periwinkles per year. Shellfish foraging areas were estimated to be 4653 m² on Koram Island

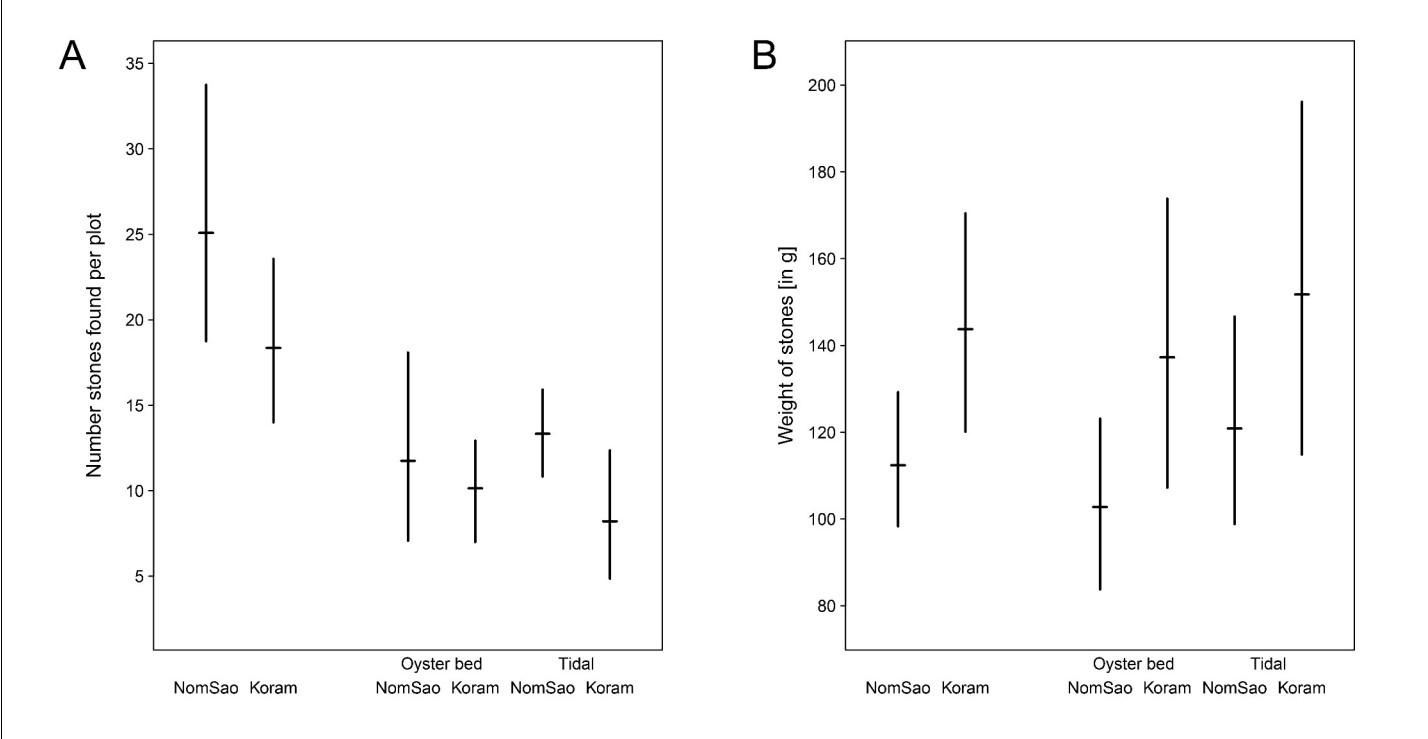

**Figure 3.** Stones available on Koram and NomSao Islands. (**A**) Weight of stones (with bootstrapped 95% confidence intervals) found on Koram and NomSao Islands, separated for oyster bed and tidal. (**B**) Average stone availability per island, separated for oyster bed and tidal (with bootstrapped 95% confidence intervals over observed plots).

DOI: https://doi.org/10.7554/eLife.23647.006

The following source data is available for figure 3:

**Source data 1.** Natural stone availability and weight.

DOI: https://doi.org/10.7554/eLife.23647.007

and 1959 m$^2$ on NomSao Island. Availability per square meter gives estimates for the current total number of periwinkles as 60,163 on Koram and 79,477 on NomSao Island. Extrapolating from the consumption data, and without prey population replenishment, then within a year the macaque group foraging on Koram hypothetically would consume more than the estimated current number of periwinkles on the island. On NomSao Island, however, each year the inhabitants hypothetically eat just over a tenth of the current periwinkle population on their island.

To assess the vulnerability of macaque prey species to foraging pressure, we merged predictions on prey resilience from published agent based models with available life history data on the two main prey species (oysters and periwinkles). The results are summarized in *Table 1*, and show that both species are easily harvested (they are clustered) but they also have planktonic dispersal, reducing reliance on local populations for replenishment. In addition, when comparing published reproductive sizes of periwinkles located in our point transects we found that 62% (N = 123) of individuals located in Koram plots were smaller than reported reproductive sizes (*Moore, 1937*). On NomSao Island on the other hand, none of the periwinkles in the plots (N = 100) were smaller than the reported reproductive size. Koram is therefore missing around a third of the expected larger and more mature periwinkles. This suggests that macaques preferably harvest larger prey individuals.

## Discussion

Macaque stone tool selection for shellfish processing differed between two closely adjacent islands in Khao Sam Roi Yot National Park, Thailand. On Koram Island, macaques selected significantly smaller stone tools than macaques on NomSao Island, despite targeting the same prey species. This

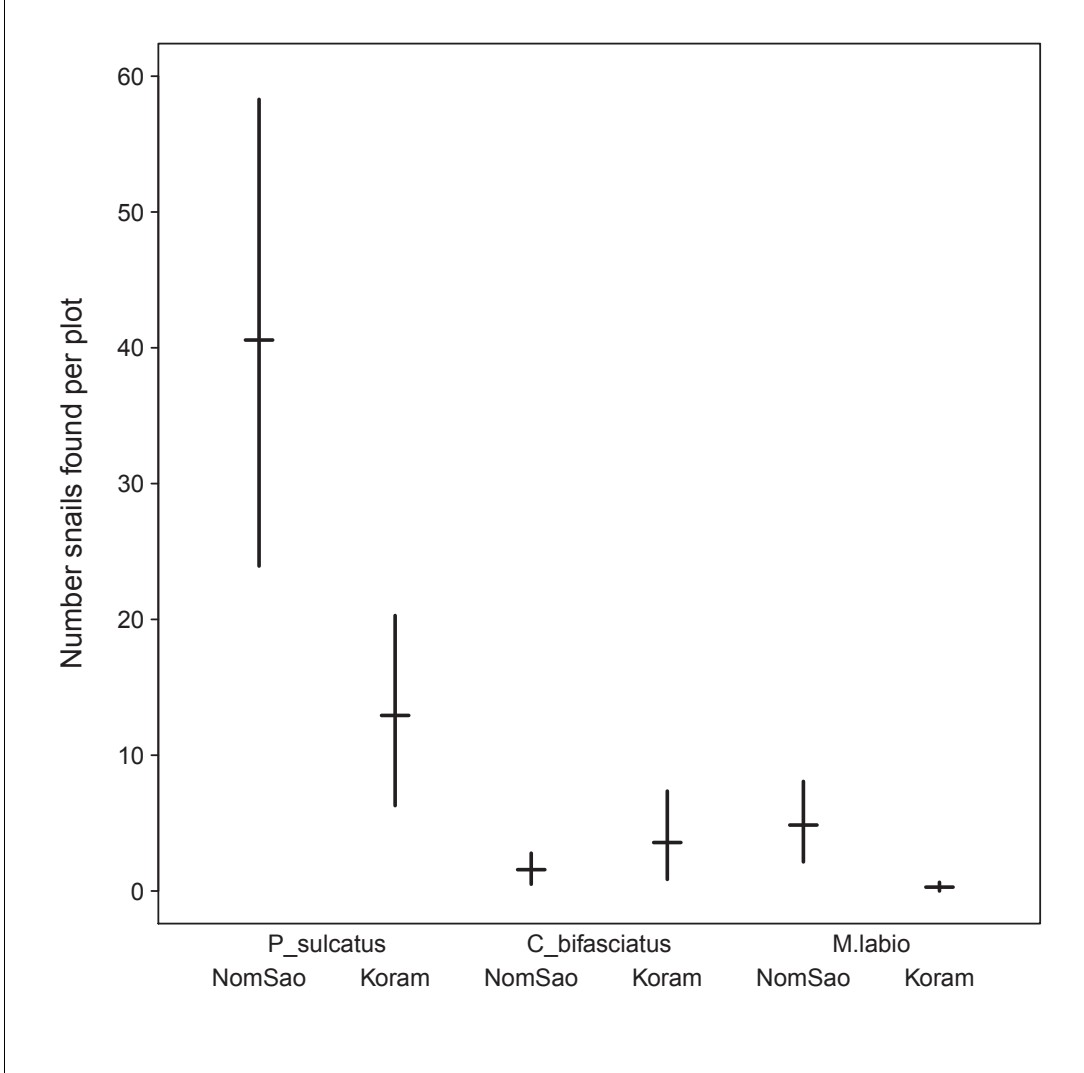

**Figure 4.** Average snail availability on Koram and NomSao Islands for three species (with bootstrapped 95% confidence intervals over observed plots).
DOI: https://doi.org/10.7554/eLife.23647.008
The following source data is available for figure 4:

**Source data 1.** Snail availability.
DOI: https://doi.org/10.7554/eLife.23647.009

pattern was not explained by available stone material on the islands, with smaller stones on average found on NomSao. The most likely factor influencing tool selection patterns were differences in prey sizes between the two islands. On Koram, the sizes of multiple prey species were significantly smaller than on NomSao, and selected tool sizes correlated positively with targeted prey size on both islands. The fact that macaques on Koram selected smaller tools to use on the shellfish, despite stones there being larger on average than on NomSao, suggests that the macaques actively select task-specific stone sizes. Our result supports a previous finding from a wild macaque population on the west coast of Thailand, where macaques selection of tool size is associated with the size of the prey (*Gumert and Malaivijitnond, 2013*).

In addition to size differences, multiple prey species on both islands were less available on Koram than on NomSao Island. Feeding pressure, as measured by rates of prey consumption, was estimated to be significantly higher on Koram Isalnd, where, unlike NomSao Island there was a dense population of tool-using predators foraging for shellfish. However, shellfish of similar size were of similar maturation stage on both Koram and NomSao Islands. This outcome suggests that there

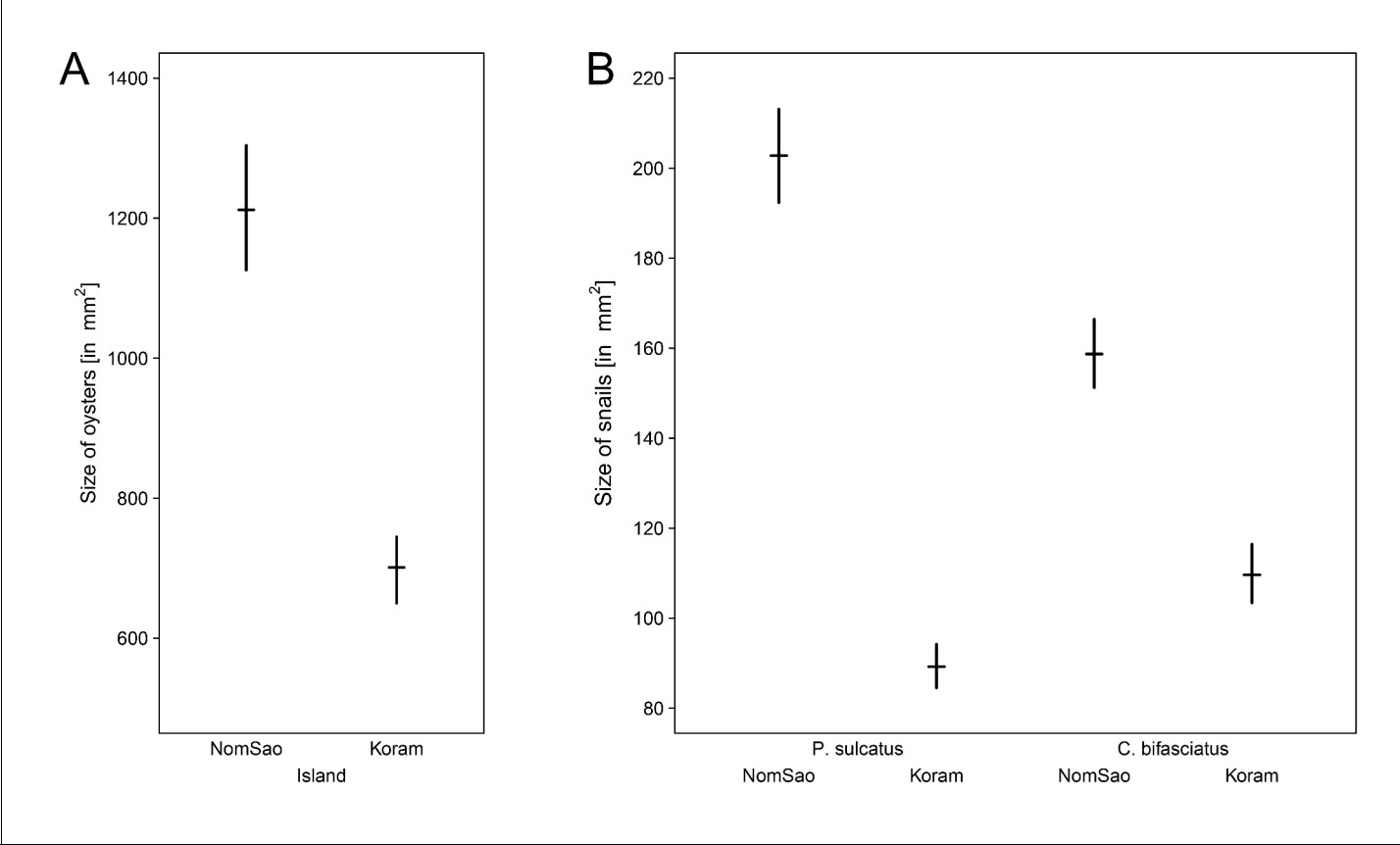

**Figure 5.** Prey size on Koram and NomSao Islands. (**A**) Average size of oysters (with bootstrapped 95% confidence intervals) found on Koram and NomSao Islands. (**B**) Average snail size (volume) (with bootstrapped 95% confidence intervals) found on Koram and NomSao Islands.

DOI: https://doi.org/10.7554/eLife.23647.010

The following source data is available for figure 5:

**Source data 1.** Oyster size.
DOI: https://doi.org/10.7554/eLife.23647.011
**Source data 2.** Snail size.
DOI: https://doi.org/10.7554/eLife.23647.012

have been no life history changes in the Koram shellfish, adapting them to higher predation levels on that island. This indicates a rather recent change in feeding intensity, meaning that the difference in prey size between the two islands likely results from a large number of macaques preferentially harvesting large prey on Koram Island, and not a long-term evolutionary response to predation pressure.

We do not know how long macaques have been on Koram Island, nor how long they have been provisioned, and range at such high population levels. Based on local reports though, we can infer that macaques have been on these islands for at least 30 years. It is therefore possible that there has been no opportunity for a long-term evolutionary relationship between this population of macaques and their prey. All others tool using macaque populations are in the Andaman Sea (*Gumert and Malaivijitnond, 2013*; *Carpenter, 1887*), so exactly how these macaques arrived in the Thai Gulf, with tool technology, remains a mystery.

Assessment of the life history of the main prey species, periwinkles and rock oysters, revealed that the both species appear in large clusters and are therefore susceptible to overharvesting. However, the number of mature individuals and their reproduction rate on the islands turned out to not be useful proxies for estimating population replenishment rates, as both species undergo planktonic reproductive stages. New prey therefore arrives on the ocean currents, with both Koram and

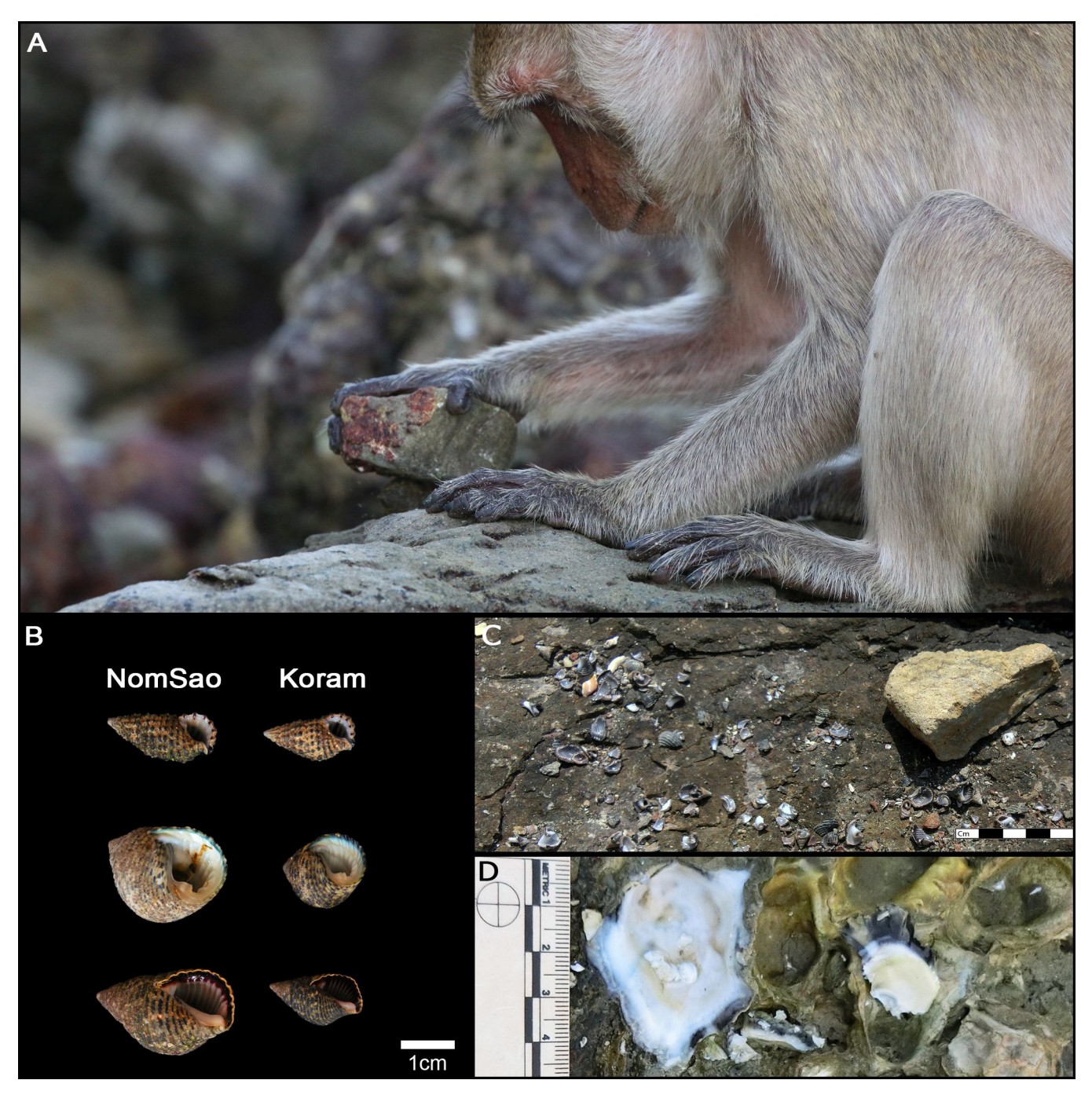

**Figure 6.** Long-tailed macaque tool use. (**A**) Adult male long-tailed macaque using a stone tool to crack open a snail. (**B**) Size difference between NomSao and Koram Islands of most commonly harvested snails. (**C**) Abandoned macaque tool at shellfish cracking site, with prey remains. (**D**) Recently harvested oysters (white) are clearly distinguishable from older oysters (grey).
DOI: https://doi.org/10.7554/eLife.23647.013

NomSao Islands receiving the same input of new individuals, at the same rate. On both islands, shellfish foraging sites (where the oyster beds are located) face north-west and the currents transporting planktonic larvae and therefore supply of new prey affect each island similarly. For periwinkles, the continual arrival of new individuals offers an explanation for why the prey population on Koram has

**Table 1.** Life history information of main prey species.

| Factors influencing population resilience | Prediction | Rock oyster | Tropical periwinkle |
|---|---|---|---|
| Resource Aggregation/Clustering | High aggregation → slow recovery | High aggregation | High aggregation |
| Size and age at sexual reproductive maturity | Large size at sexual maturity → slow recovery | Large: first year grows 25 mm. Able to reproduce in the first year | Small: first year grows 14 mm (17.44 mm second year). Reproduced in second year |
| Reproductive output (per individual) | High reproductive output → fast recovery | High (50 to 200 million) | Low (10.000–100.000) Only 2% survive until sexual maturity |
| Larvae stage | Attached → slow recovery | Unattached: Planctonic larvae | Unattached: Planctonic larvae |

DOI: https://doi.org/10.7554/eLife.23647.014

not been entirely consumed, despite high foraging rates, as population replacement is not dependent on local mature individuals.

Other environmental factors are rather unlikely to have caused a reduction in shellfish size. Sources of shellfish harvesting other than macaques cannot be entirely excluded, but are minimal. For example, shorebirds (Scolopacidae) have been occasionally observed foraging on Koram and Nom-Sao Islands, but these birds primarily eat crustaceans (*Moreira, 2008*), and are solitary foragers with negligible effect on prey numbers. Both islands are also visited regularly by tourists and locals. On NomSao, we have never observed locals to harvest shellfish, as the island is considered holy and removing anything from there is unacceptable. On Koram Island, seafood harvesting was only observed once during a full year of eight hour daily focal follows. The target prey of this gatherer was exclusively pen shells (*Atrina* sp.), which are not targeted by the monkeys. Tourists have not been observed harvesting molluscs on either island.

Bringing together the observed macaque behavior and environmental conditions, we interpret the variation seen in macaque stone tool selection and shellfish characteristics on Koram and Nom-Sao Islands as the result of a feedback loop driven by the level of predation pressure. The inverse correlation between the number of tool-using macaques, prey size and availability, suggests that predation pressure may be the primary cause of shellfish body size and population on these islands. Particularly, when a population of macaques becomes densely populated, the effect on their prey population becomes obvious.

If our conclusions are correct, we interpret the current size distribution of shellfish on Koram as an indication that the macaques are in the process of reducing prey numbers and size which ultimately may lead to unsustainable resource exploitation. This situation could result from tool use being a relatively recent phenomenon on the island, occurring too fast to allow for evolutionary responses from prey species. Alternatively, group sizes on Koram may have only recently reached densely populated levels, likely through increased provisioning, causing an exertion of greater pressure on the shellfish population. These alternative explanations might be testable through archaeological excavation (*Haslam et al., 2016b*), for example we might expect to find larger shellfish remains and tool evidence on Koram Island as we go further back in time, with the size approaching that seen today on NomSao Island. This approach would also assist comparisons with human archaeological records of shellfish over-exploitation.

Macaques have been reported to impact the local flora and fauna in areas where they are provisioned and densely populated (*Gumert, 2011*). However, comparative data from other primates that forage on intertidal resources (*Malaivijitnond et al., 2007*; *Hall, 2009*; *Fernandes, 1991*) are lacking. Away from the coastlines, there is evidence that non-human primates can hunt prey at unsustainable levels, for example wild chimpanzees (*Pan troglodytes schweinfurthii*) at Ngogo in Uganda hunt red colobus monkeys (*Procolobus rufomitratus*) at a rate that may lead to local extinction of the latter (*Teelen, 2008*).

Similar to the effects posited for human coastal exploitation, our results suggest that tool-assisted shellfish consumption by a densely populated non-human primate species might lead to unsustainable foraging. Over-harvesting could ultimately lead to the loss of technological knowledge in these

macaques. With the decline of prey species, the benefit from using stone tools would decrease, leading to less tool use or its cessation. To definitively answer the question of whether continued shellfish depletion might lead to such a loss at Koram Island, we will need to monitor future developments. However it unfolds, these macaques provide an interesting case for potential feedback systems between a technological predators and its prey. The isolated nature of island-dwelling macaques therefore makes them a useful model for assessing how simple technologies may impact prey population dynamics, life histories, and phenotypes.

## Materials and methods

### Field site

Koram (N12°14′32", E100°0′34") and Nom Sao (N12°13′51", E100°0′17") Islands are located in Khao Sam Roi Yot National Park (KSRY), Prachuap Khiri Khan province, Thailand. Both islands are arid and contain no freshwater bodies, with limestone karst interiors that are covered with dwarf evergreens and deciduous scrub flora (*Figure 1*). Koram Island is located about 1 km offshore from Sam Roi Yot beach on the Thai eastern mainland. It is approximately 0.45 km$^2$, with 3.5 km of coastline comprising of limestone cliff shore on the side of the island facing the open gulf, and rocky shores and sandy beaches on the side of the coast facing the mainland. NomSao Island is 0.37 km southwest of Koram, with an area of 0.10 km$^2$, including 1.32 km of coastline. The shoreline is not completely accessible for the monkeys as steep overhanging cliffs make some parts unsuitable for foraging. We measured the accessible shoreline on which the monkeys were able to forage by taking GPS points at the furthest edge of suitable foraging zones, and we calculated the total length of suitable shoreline using Google Earth.

The macaque groups on Koram and NomSao Islands are habituated to the presence of humans as a result of provisioning by locals and tourists with food and water. However, they also forage naturally using percussive stone technology. The macaques process predominantly sessile rock oysters (*Saccostrea cucullata*), and the main gastropod species processed are the tropical periwinkle (*Planaxis sulcatus*). They also harvest bifasciated cerith (*Clypeomorus bifasciatus*), and tooth-lipped snail (*Monodonta labio*). Other less commonly processed food species are conches, various bivalves and crabs, as well as coconuts that occasionally wash ashore, and the dried seeds of provisioned mangoes. At the end of our first data collection period in December 2014, the studied Koram macaque group contained 64 individuals (about 178 ind/km$^2$). Twenty-five (12 ♂; 13 ♀) of 36 adult and adolescent macaques (i.e. > 4 years) were tool users, or 69.4% of the group, and 3 of the 14 (21.4%) juveniles (i.e. 1–4 years) were observed using tools. A smaller group of about 10 individuals is usually excluded from the coast by the larger group and less easily observed, but they also forage on shellfish with tools when they are able to access the shores. On NomSao, there are at least nine adult males that have been observed and individually identified (approximately 90 ind/km$^2$) with at least four being tool users. There are no females or juveniles on NomSao, and it is likely that the males there were dispersed from Koram. Macaques live in multi-male multi-female groups where males emigrate from the natal group when they are mature (so-called male dispersal), which is believed to reduce the inbreeding depression in the populations.

### Data collection

We collected data over two field seasons, from May to December 2014, and from September to October 2015. During the first season, we collected tools from the Koram group directly after observing individuals using them as either oyster or snail foraging tools. In the second field session, we focused on behavioral observations and tool collection on the NomSao group, as well as the sampling of prey size, and food and stone availability on both islands.

We collected used stone tools from Koram and NomSao Islands. Since the NomSao group contained only adult males, we limited our analyses on Koram to tools used by adult males, only to keep tool dimensions comparable between sites, as males are known to use larger tools (*Gumert and Malaivijitnond, 2013*). Tools on Koram were collected only after we observed male monkeys using and discarding them, to prevent female choices appearing in our data set. On NomSao, we additionally collected abandoned tools from recent cracking sites as only adult males could have used them on this island. Stones at recent cracking sites were identified as tools if they were

conspicuously placed on a boulder, accompanied by food debris on or adjacent to the stone, and/or showed use-wear marks from being struck. For each tool collected, we recorded whether oysters or snails were processed (identified through prey remains on the tool), and weighed the stone on a portable digital scale. We measured the length, width and depth by photographing the stone in line with a ruler. Where possible, we also noted the identity of the tool user. Our final data set included 93 oyster tools from 14 individuals on Koram and 93 oyster tools from 4 individuals on NomSao as well as 45 snail tools from 9 individuals on Koram and 22 snail tools from 4 individuals on NomSao.

We used point transects to sample the size and availability of stones between Koram and Nom-Sao Islands. We applied 14 point transects at 7 locations spaced 100 m apart along the shores of each island. At each location, we demarcated two 20 × 20 cm plots, one located in the lower littoral zone along the water line where snails were abundant, and one located higher up in the littoral zone amongst the oyster beds. We counted, weighed, and measured the lengths and widths of all stones found within each point transects. We excluded stones that were 20% smaller than the smallest tool used by the macaques on Koram. The upper limit of potential stone tools was set as the size of the plot (max = 20 cm$^2$). However, no stones found in the plots reached this size.

To compare the ecological conditions between the two islands, we collected data on prey size and tool availability. To compare the sizes of oysters exploited by the macaques, we traversed the oyster beds of both islands, and measured the length and width of recently cracked oysters. From these measurements, we calculated the size of 345 oysters on NomSao and 673 oysters on Koram. Recently-cracked oysters were identified by the presence of flesh remains on the opened valve, or the white shiny inner surface of the opened valve (older oysters are rapidly discoloured by sea water). We did not carry out plot sampling for oyster availability as oyster beds were distributed continuously along the lengths of rocky shores of both islands and therefore were not a limiting foraging factor.

To compare snail availability and size between Koram and NomSao, we applied 14 systematic point transects on each island. Snails were more abundant in the lower littoral zones and thus exposed during the low tide, so we conducted sampling during the low tide hours of each day. During these low tide hours, when the lower littoral zone was exposed, we demarcated 1 × 1 m plots at each point transect along the shore in a line, 100 m apart from one another. As a measure of snail availability, we counted all snails of the three most commonly processed snail species (*Planaxis sulcatus*, *Clypeomorus bifasciatus,* and *Monodonta labio*) in each plot. On Koram Island, we found a total of 236 (*P.sulcatus* = 181; *C.bifasciatus* = 50; *M.Labio* = 4) snails and on NomSao Island a total of 658 (*P.sulcatus* = 568; *C.bifasciatus* = 22; *M.Labio* = 68) snails in all point transects combined. To obtain data on snail size, we used calipers to measure the maximum shell length and width of 100 randomly selected individual snails of each species and on each island, and we weighed each snail on a portable digital scale (1–1000 g).

To further investigate whether the underlying reason for observed size differences between the three main marine snail prey species stems from overharvesting large individuals, or is rather a result of an evolutionary change in snail biology, we collected a representative sample of four different size categories of each species on each island (N = 77). We assessed the maturation stage of each size category by investigating the developmental stage of the reproductive organs. The samples were stored in 70% ethanol and their sexual organs were inspected under a microscope after breaking their outer shell. Maturation was classified into four developmental stages: Immature (no traces of reproductive organs); Subadult (testis/ovaries occupy less than 50% of the upper whorl and are therefore not yet able to reproduce); Mature (testis/ovaries occupy more than 50% but less than 80% of the upper whorl, these samples might be able to reproduce); and Completely Mature (testis/ovary occupy more than 80% of the upper whorl). We aimed at collecting four different snail sizes (size equivalent for each island), however, since *Monodonta labio* was very rare on Koram we were not able to find any large specimens of that species and therefore excluded it from the comparative analysis.

To investigate the sustainability of the marine prey consumption, we combined published information on the life history of our main prey species with the outcome of an agent-based model (*Morrison and Allen, 2017*), concentrating on the main prey species (*P. sulcatus, C. bifasciatus,* and *M. labio*). These species have a reproductive system which includes a planktonic stage of the larvae that can disperse great distances by water currents before settlement.

Observational data were collected on the proportions of foraging time that the macaques engaged in tool-aided foraging, and on their rates of tool use. During focal sampling, all individuals were sampled in random order on continuous rotation, for a five minute duration each time. During foraging, we recorded the type of food item processed, noting whether the subject was cracking a sessile oyster or unattached gastropod, and recording the species whenever possible. We counted the number of prey items consumed per focal observation time of 5 min and conservatively estimated an average foraging time of 4 h (during low tide) per individual to extrapolate the number of shellfish items eaten per day (and per year) on Koram Island. We used the amount of daily prey consumption on Koram Island to estimate the total foraging pressure that one tool-using macaque can place on the prey population. We used this same value to estimate foraging pressure for the neighbouring NomSao population, for which long-term focal observations were not available (Tan and Luncz, personal communication). We then used our data on snail abundance per island to extrapolate the potential time needed to deplete the existing prey population. For that calculation we multiplied the length of suitable coastal foraging areas per island with an average width of foraging grounds of 3 m. We used the resulting area and the number of snails found per surveyed square meter to estimate the total number of snails available on each island.

## Statistical analyses
### Tool choice
To analyze whether the weights of tools selected to crack open oysters differed between the two observed populations we used a linear mixed model (LMM) (*Baayen, 2008*). In this model, we included the population as fixed effect and individual ID as random effect. As response we used the weight of the stone tools. Prior to running the model, we square root transformed the response variable to achieve a more symmetrical distribution.

We also tested whether the weight of tools used to crack open snails differed between the two populations. As we had no information on the individuals that used the stones, we ran a linear model (LM). Into this model, we included the population as fixed effect. As response we used the weight of the stone tools. To achieve a more symmetrical distribution, we log transformed the response variable prior to running the model.

To test whether the size of a given maturity stage (immature or mature) differed between the two populations we ran a linear model (LM) with the size category (*Barnosky et al., 2011*; *Jackson et al., 2001*; *Small and Nicholls, 2003*; *Mannino and Thomas, 2002*) as response. As predictors we included the prey species into the model as well as the two-way-interaction between island and snail maturation stage.

To increase the confidence of the linear mixed model and the linear model analyses, we ran a permutation test in which we randomized the assignment to the island for each individual. We did this 10000 times and compared the distribution of the revealed permutation results with the original test statistic to determine p-values.

For all models, we checked whether the assumptions of normally distributed and homogeneous residuals were fulfilled by visually inspecting a qqplot and the residuals plotted against fitted values. In both models, we found no obvious deviations from these assumptions.

For the LMM, we additionally checked for model stability by excluding each individual at a time from the data. A comparison of the model estimates derived for the reduced data with those derived by the full data set indicated no influential cases to exist. For the LM, we tested model diagnostics using the R functions 'dffits,' 'dfbeta,' and 'cooks.distance', and we additionally checked for leverage and did not find any assumptions violated.

The p-values for the fixed effects were based on a likelihood ratio test (LRT), comparing the full with the model reduced by the fixed effect (*Dobson and Barnett, 2002*; *Barr, 2013*) using the R function anova with argument test set to 'Chisq' and in case of the LM's to 'F'. To allow for a LRT, we fitted the LMM using Maximum Likelihood (rather than Restricted Maximum Likelihood) (*Bolker et al., 2009*). For the third model (investigating snail maturation), we tested for the significance of our full model using a LRT by comparing the full with the null model (model reduced by all fixed effect).

The models were implemented in R (version 3.2.3) (*R Developing Core Team, 2010*). The LMM was fitted using the function lmer of the R package lme4 (*Bates and Maechler, 2010*) and the LM using the function lm.

## Impact on prey

To compare oyster size between the two islands, we bootstrapped the measured oyster sizes for each island 1000 times and compared the confidence intervals at the level of 95% to each other. To test for differences in snail size between the islands, we conducted the same procedure for two snail species (*C. bifasciatus* ($N_{NamSao}$ = 101, $N_{Koram}$ = 119); *P. sulcatus* ($N_{NamSao}$ = 100, $N_{Koram}$ = 123)). To compare the availability of the different snail species between islands for each species individually, we bootstrapped the number of snails we found on each transect 1000 times and compared the confidence intervals at the level of 95% between the islands.

To compare stone availability and stone size separated for island and location we bootstrapped i) the number, and ii) the weight of the stones we found per transect 1000 times and compared the confidence intervals at the level of 95% between the islands. For each test, we additionally applied two sample t-tests. All bootstraps and t-tests were implemented in R (version 3.2.3).

## Acknowledgements

The National Research Council of Thailand permitted LVL, AT, MH, and MG to conduct research in Thailand, and the Thai Department of National Parks, Wildlife, and Plant Conservation gave permission to enter and conduct research in Khao Sam Roi Yot National Park. We thank Chirasak Sutcharit from Chulalongkorn University for the evaluation of the marine snail specimens. We thank Dr Nate Dominy, Nalina Aiempichitkijkarn and four reviewers for helpful comments on the manuscript and our field assistants Lauren O'Boyle and Magdalena Svensson for help with the data collection. This research was funded by European Research Council Starting Grant no.283959 (PRIMARCH) awarded to MH. During writing LVL was funded by a Leverhulme Trust Research Grant.

## Additional information

### Funding

| Funder | Grant reference number | Author |
| --- | --- | --- |
| European Research Council | 283959 | Michael Haslam |
| Leverhulme Trust | | Lydia V Luncz |

The funders had no role in study design, data collection and interpretation, or the decision to submit the work for publication.

### Author contributions

Lydia V Luncz, Conceptualization, Data curation, Formal analysis, Investigation, Methodology, Writing—original draft; Amanda Tan, Data curation, Writing—original draft; Michael Haslam, Funding acquisition, Writing—original draft; Lars Kulik, Formal analysis, Writing—original draft; Tomos Proffitt, Conceptualization, Writing—original draft; Suchinda Malaivijitnond, Conceptualization, Resources, Validation, Project administration; Michael Gumert, Conceptualization, Supervision, Validation, Investigation, Methodology, Writing—original draft

### Author ORCIDs

Lydia V Luncz, http://orcid.org/0000-0003-2972-4742
Michael Haslam, http://orcid.org/0000-0001-8234-7806

### Decision letter and Author response

Decision letter https://doi.org/10.7554/eLife.23647.017
Author response https://doi.org/10.7554/eLife.23647.018

## Additional files

**Supplementary files**

• Source data 1. Koram Island shellfish foraging. The number of prey items consumed during behavioral observation of daily shellfish foraging on Koram Island, Thailand.
DOI: https://doi.org/10.7554/eLife.23647.015

• Source data 2. Maturation stages. The size and maturation stages of the main prey species harvested by tool using macaques on Koram and NomSao Island, Thailand.
DOI: https://doi.org/10.7554/eLife.23647.016

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
