## [Decision Letter]

Thank you for submitting your article "Resource depletion through primate stone technology" for consideration by *eLife*. Your article has been favorably evaluated by Andrew King (Senior Editor) and four reviewers, one of whom is a member of our Board of Reviewing Editors. The following individual involved in review of your submission has agreed to reveal their identity: Thibaud Gruber (Reviewer #3).

The reviewers have discussed the reviews with one another and the Reviewing Editor has drafted this decision to help you prepare a revised submission.

Summary:

All reviewers strongly agreed that the subject of the manuscript – the idea that tool-assisted shellfish over-exploitation by non-human primates may be affecting prey size and abundance – and the suggestive results presented in the manuscript are of very high interest. However, there are several larger concerns/ limitations of the study that we are uncertain can be addressed adequately in a revision in reasonable time (2 months, and without new data collection). We collectively decided to provide you with an explanation of the major issues that would need to be satisfactorily addressed prior to publication in *eLife*, along with other comments (see below) that you might want to consider for your revision to *eLife* or your next resubmission to another journal. If you feel that these issues can be fully addressed within two months, then we look forward to receiving and reviewing your revised manuscript, given our enthusiasm about the topic.

Essential revisions:

1) Definitive insight into resource use intensity differences between the two islands would be required. Presently, the main argument for a difference is based on the presence of 80 macaques on Koram and 9 on NomSao, leading to inference of higher resource use intensity on Koram. However, Koram appears much larger (although information on island size is not provided), and with more available coastline. Thus, resource use intensity might not be that different between the islands?

1A) One suggestion from reviewers to help address this issue is to consider incorporating life-history models of prey taxa (based on prey size) to determine if the current harvesting rates surpass the maximum sustainable yield. See Morrison and Allen (2015, Agent-based modelling, molluscan population dynamics, and archaeomalacology, Quaternary International) and Rick et al. (2016, Millennial-scale sustainability of the Chesapeake Bay Native American oyster fishery, PNAS). Note that this analysis would assume that the size-age relationship is constant and there has not been an evolutionary response, which should be stated (see related comment #4 below).

1B) Another suggestion from reviewers is that perhaps the individual macaque foraging data could be used to create a rough estimate of total macaque foraging per island divided by available coastline, to infer whether resource pressures are actually substantially different? This analysis has its own limitations, but along with the suggestion above it might provide a better baseline setup for the main premise of the paper than what is currently available.

2) The argument for similar ecological context between the two islands must be dealt with much more rigorously than in the present version of the manuscript, in which the case is based largely on geographic proximity. Questions that need to be addressed include differences in human presence/disturbance between the islands (including potential human shellfish harvesting), soil property differences, oceanic current, alternative predator presence and abundance (e.g., sea birds). All of these (and other) ecological differences could impact shellfish growth or mortality rates.

3) In part because of the above issues, but also because of the general limitations of the dataset (e.g., that the authors are making n=1 comparison between two islands), the general conclusions need to be tempered throughout the manuscript. Related to this comment, there are several unsubstantiated claims in the manuscript that should be corrected (or evidence strongly supporting these conclusions, provided). In the Abstract, the authors mention that this study demonstrates "profound effects on the reproductive biology of shellfish," but this is not shown in the paper. In the Discussion, the statement that "one snail species had almost gone extinct on Koram Island" needs to be removed or edited, unless more thorough ecological data and analysis are provided to support this claim.

4) The topic of the mechanism of size change should be addressed more directly within the paper (in terms of the possibilities, what is shown/ can be known based on the data, and what the authors hypothesize). Human over-exploitation can result in a change in size by three major pathways: 1) an evolutionary change in size due the directional selection caused by size selective harvesting, 2) a younger age structure (with more young and therefore smaller individuals) due to increased mortality by harvesting, 3) evolution of a faster life-history strategy caused by the increased mortality which results in a younger age structure independently of harvesting mortality. It would be interesting and make for a stronger argument for the similarity between macaque over-exploitation and human overexploitation if the source of size difference could be more precisely defined. Is it due to a change in age structure? Are macaques selective in their choice of prey size?

5A) Statistical analyses. The reviewers request the incorporation of permutation or similar analyses to complement the linear mixed model analyses – i.e., randomizing the observed values for one variable being considered in a particular test across the whole sample, computing the test statistic in the same way it was computed in the original test, repeating that step 10,000 times, and then considering the original test statistic against the distribution of permutation results as an empirical p-value. Providing such a secondary analysis would help increase confidence in the robustness of the results.

5B) Statistical analyses. The authors conclude that there is tool choice, but the distribution of used tools and available stones on Nom Sao is very similar and random stone choice would probably result in the same distribution of stone use. A proper resource selection analysis for tool use instead of simple comparison of means would likely make for a stronger argument for tool size selection (see Manly et al. (2002) Resource Selection by Animals: Statistical Design and Analysis for Field Studies). Data collection for available stone tools excluded stones that were >=20% smaller than smallest used tool. Were stones >=20% bigger than biggest tool also excluded? Finally, can the authors address the apparent inconsistency between the size distribution of used vs. available stones on Koram? i.e., is the sample of available stones representative?

6) Data availability. There is no mention of data availability in the current version of the manuscript. This issue needs to be addressed, with all underlying data, including the full database of tool and shellfish sizes, made available through appropriate repositories without restriction.

7) Improvement to the figures (other than Figure 1 and Figure 6) is necessary.

[Editors' note: further revisions were requested prior to acceptance, as described below.]

Thank you for resubmitting your work entitled "Resource depletion through primate stone technology" for further consideration at *eLife*. Your article has been favorably evaluated by Andrew King (Senior Editor) and four reviewers, one of whom is a member of our Board of Reviewing Editors.

The manuscript has been improved but there are some remaining issues that need to be addressed before acceptance, as outlined below. Given that this has been reviewed several times now and remains in an unacceptable form, we are prepared to offer only one more opportunity to provide an acceptable version of the manuscript.

In the manuscript, the authors explore the impact of tool-assisted predation by long-tailed macaques on morphological and life-history trait on prey species (mollusc). In this revision, the reviewers all agreed that the authors now provide more support for their hypothesis that the molluscs are subject to over-harvesting pressure by the tool-using macaques, with a potential feedback loop with tool size. The results are still viewed as preliminary in some ways, but consideration for publication in *eLife* is warranted given the strengths of the inferences made thus far, the fundamental interest in the topic, and the high relevance of this work to considerations of the impacts of human over-exploitation on harvested species' population dynamics, life histories, and phenotypes.

However, three major issues of consensus were raised by reviewers that must be addressed before the manuscript can be considered further.

Necessary revisions:

1) A strong effort needs to be made to rework the manuscript to reflect the points previously raised by reviewers regarding the over-simplistic assumptions of the study design as presented in the original submission and to more holistically account for the expanded analyses. The revised manuscript appears to fit new analyses into the previous framework, when a new presentation is needed. This results in some confusion/disconnect between the too-over-simplified motivating setup and a less effective/clear manuscript overall than what is possible. Both the Introduction and the Discussion, especially, need to be wholly reconsidered and reorganized, rather than lightly edited.

2) The evidence that the overharvesting of prey species is unsustainable is not viewed as sufficiently supported to justify some of the strong conclusions made by the authors on this point. Specifically, the authors compare the situation at Koram to the likelihood of population resilience according to the literature (Morisson and Allen 2017), rather than performing an actual viability analysis. The authors themselves write: "The number of adult animals and their reproductive output on the islands therefore is not a valid proxy to estimate sustainability of the prey species". Thus, please adjust discussion of these results in all cases to more carefully express appropriate caution (e.g., following the approach used in the current version of the abstract, which strikes a good balance).

3) The estimate of how long before macaques could consume all periwinkles on the two islands, needs revision/clarity. If periwinkle snails are able to reproduce at the second year, this has to be accounted for in the prediction of the depletion, particularly for NomSao island.

---

## [Author Response]

Summary:

All reviewers strongly agreed that the subject of the manuscript – the idea that tool-assisted shellfish over-exploitation by non-human primates may be affecting prey size and abundance – and the suggestive results presented in the manuscript are of very high interest. However, there are several larger concerns/ limitations of the study that we are uncertain can be addressed adequately in a revision in reasonable time (2 months, and without new data collection). We collectively decided to provide you with an explanation of the major issues that would need to be satisfactorily addressed prior to publication in eLife, along with other comments (see below) that you might want to consider for your revision to eLife or your next resubmission to another journal. If you feel that these issues can be fully addressed within two months, then we look forward to receiving and reviewing your revised manuscript, given our enthusiasm about the topic.

We would like to thank the editor and the reviewers for the detailed and helpful comments to our manuscript. In order to address the concerns of the reviewers and to support or findings we have added three new lines of evidence. These consist of:

1) Inflicted foraging pressure on prey species, estimated using focal observation during foraging events.

2) The likelihoods of population resilience of the main prey species (rock oyster and tropical periwinkle) using agent based models and published data on the life history of these species.

3) Snail size and their respective maturation stages to analyse size-age relationship of prey species.

The outcomes of this new data supports our previous conclusion that prey species are harvested in an unsustainable manner through tool assisted foraging by long tailed-macaques.

Essential revisions:

1) Definitive insight into resource use intensity differences between the two islands would be required. Presently, the main argument for a difference is based on the presence of 80 macaques on Koram and 9 on NomSao, leading to inference of higher resource use intensity on Koram. However, Koram appears much larger (although information on island size is not provided), and with more available coastline. Thus, resource use intensity might not be that different between the islands?

We previously reported only the island sizes, and have now included data to provide the reader with a better picture of foraging sites. On both islands, only parts of the shoreline are suitable for shellfish foraging or even accessible to the monkeys (due to steep cliffs). We have provided new data on the exact locations (through GPS coordinates) where the monkeys are able to access the coast for shellfish foraging; using Google Earth we then measured the length of the available shoreline on each island.

Additionally, we divided the available shoreline by the number of tool users foraging on each island, and estimated the predation pressure caused by the monkeys on their respective islands (for details please see our reply to #4 below).

1A) One suggestion from reviewers to help address this issue is to consider incorporating life-history models of prey taxa (based on prey size) to determine if the current harvesting rates surpass the maximum sustainable yield. See Morrison and Allen (2015, Agent-based modelling, molluscan population dynamics, and archaeomalacology, Quaternary International) and Rick et al. (2016, Millennial-scale sustainability of the Chesapeake Bay Native American oyster fishery, PNAS). Note that this analysis would assume that the size-age relationship is constant and there has not been an evolutionary response, which should be stated (see related comment #4 below).

We used the outcomes of the models described by Morrison and Allen to evaluate the influence of the life history of the main target species (oysters and periwinkles) on the reproductive efficiency and potential impact on prey decline. We used existing literature on the life history of these species to investigate prey population resilience and the likelihood of overharvesting by tool using macaques.

The criteria and prey species information we used included: clustering, age and size of sexual maturity, reproductive output, and larvae stage. We summarize the results in the new Table 1.

We compared our existing data on the average body size of prey species on both islands to the size of general maturation reported in the available literature. We also now include an estimate of the percentage of prey animals that were below the reproductive size.

1B) Another suggestion from reviewers is that perhaps the individual macaque foraging data could be used to create a rough estimate of total macaque foraging per island divided by available coastline, to infer whether resource pressures are actually substantially different? This analysis has its own limitations, but along with the suggestion above it might provide a better baseline setup for the main premise of the paper than what is currently available.

Long term observational data – one year of full-day focal follows – is available only for Koram Island, although shellfish foraging is a daily activity on both islands during low tide (4h per day). We used individual focal foraging data to calculate the pressure that one tool-using macaque exerts on the prey population on Koram. We counted the number of prey items consumed per 5 minutes of focal observation by every tool-using monkey on Koram, and used the average low tide foraging time of 4 hours per individual to extrapolate the number of items eaten per day and per year. We used the same data to estimate the number of prey items eaten per day per individual In the NomSao Island population, which have similar foraging times to individuals on Koram (Tan and Luncz, personal communication). Finally, we combined this information with our existing data on snail abundance per island to calculate the potential time needed to deplete the existing prey population.

*2) The argument for similar ecological context between the two islands must be dealt with much more rigorously than in the present version of the manuscript, in which the case is based largely on geographic proximity. Questions that need to be addressed include differences in human presence/disturbance between the islands (including potential human shellfish harvesting), soil property differences, oceanic current, alternative predator presence and abundance (e.g., sea birds). All of these (and other) ecological differences could impact shellfish growth or mortality rates.*

We have included more information on the main factors that could influence shellfish abundance, including human shellfish harvesting and disturbance, soil properties, oceanic current, and alternative predators (such as sea birds).

3) In part because of the above issues, but also because of the general limitations of the dataset (e.g., that the authors are making n=1 comparison between two islands), the general conclusions need to be tempered throughout the manuscript. Related to this comment, there are several unsubstantiated claims in the manuscript that should be corrected (or evidence strongly supporting these conclusions, provided). In the Abstract, the authors mention that this study demonstrates "profound effects on the reproductive biology of shellfish," but this is not shown in the paper. In the Discussion, the statement that "one snail species had almost gone extinct on Koram Island" needs to be removed or edited, unless more thorough ecological data and analysis are provided to support this claim.

We have added new data to the manuscript comparing the reproductive stages of snail species between the islands. Additionally we included information the life history of the main target prey to better demonstrate the effects that tool-using monkeys have on shellfish reproductive biology. We have also edited the manuscript to avoid over-generalised and unsubstantiated claims.

4) The topic of the mechanism of size change should be addressed more directly within the paper (in terms of the possibilities, what is shown/ can be known based on the data, and what the authors hypothesize). Human over-exploitation can result in a change in size by three major pathways: 1) an evolutionary change in size due the directional selection caused by size selective harvesting, 2) a younger age structure (with more young and therefore smaller individuals) due to increased mortality by harvesting, 3) evolution of a faster life-history strategy caused by the increased mortality which results in a younger age structure independently of harvesting mortality. It would be interesting and make for a stronger argument for the similarity between macaque over-exploitation and human overexploitation if the source of size difference could be more precisely defined. Is it due to a change in age structure? Are macaques selective in their choice of prey size?

To address these questions we returned to the field and collected size samples for each snail prey species. Three different sizes of each snail species (N=77 in total) were examined by a marine biologist and expert in sexual organs (Dr Chirasak Sutcharit, Chulalongkorn University, Bangkok) to identify their maturation stages. We then compared the size and maturation stages between the two islands using a linear model.

5A) Statistical analyses. The reviewers request the incorporation of permutation or similar analyses to complement the linear mixed model analyses – i.e., randomizing the observed values for one variable being considered in a particular test across the whole sample, computing the test statistic in the same way it was computed in the original test, repeating that step 10,000 times, and then considering the original test statistic against the distribution of permutation results as an empirical p-value. Providing such a secondary analysis would help increase confidence in the robustness of the results.

We included the proposed permutation test into the linear mixed model as well as into the linear model analysis. The results were similar (LMM) and identical (LM) to our previous results.

5B) Statistical analyses. The authors conclude that there is tool choice, but the distribution of used tools and available stones on Nom Sao is very similar and random stone choice would probably result in the same distribution of stone use. A proper resource selection analysis for tool use instead of simple comparison of means would likely make for a stronger argument for tool size selection (see Manly et al. (2002) Resource Selection by Animals: Statistical Design and Analysis for Field Studies). Data collection for available stone tools excluded stones that were >=20% smaller than smallest used tool. Were stones >=20% bigger than biggest tool also excluded? Finally, can the authors address the apparent inconsistency between the size distribution of used vs. available stones on Koram? i.e., is the sample of available stones representative?

We have added references and discussion on tool selectiveness in long-tailed macaques (e.g., Gumert and Malaivijitnond 2013 ‘Long-tailed macaques select mass of stone tools according to food type’ Phil Trans R Soc B 368:20120413). We rephrased our results to highlight that used tools reflect tool availability on NomSao Island, and extended the Materials and methods section to clarify data collection protocols.

6) Data availability. There is no mention of data availability in the current version of the manuscript. This issue needs to be addressed, with all underlying data, including the full database of tool and shellfish sizes, made available through appropriate repositories without restriction.

We have included source data files displaying the raw data used for our main analysis: comparisons of tool size, snail and stone surveys, focal observations of foraging on Koram Island, and the maturation stages of snails.

7) Improvement to the figures (other than Figure 1 and Figure 6) is necessary.

We improved the quality of the figures as well as the labelling on the axes.

[Editors' note: further revisions were requested prior to acceptance, as described below.]

Necessary revisions:

1) A strong effort needs to be made to rework the manuscript to reflect the points previously raised by reviewers regarding the over-simplistic assumptions of the study design as presented in the original submission and to more holistically account for the expanded analyses. The revised manuscript appears to fit new analyses into the previous framework, when a new presentation is needed. This results in some confusion/disconnect between the too-over-simplified motivating setup and a less effective/clear manuscript overall than what is possible. Both the Introduction and the Discussion, especially, need to be wholly reconsidered and reorganized, rather than lightly edited.

We appreciate the observation that our new results were not adequately integrated in the previous version of our paper. We have made extensive revisions to ensure that the Introduction and Discussion logically follow from and contextualise our data. We have also ensured that the Results are better organised to allow readers to easily follow our arguments.

2) The evidence that the overharvesting of prey species is unsustainable is not viewed as sufficiently supported to justify some of the strong conclusions made by the authors on this point. Specifically, the authors compare the situation at Koram to the likelihood of population resilience according to the literature (Morisson and Allen 2017), rather than performing an actual viability analysis. The authors themselves write: "The number of adult animals and their reproductive output on the islands therefore is not a valid proxy to estimate sustainability of the prey species". Thus, please adjust discussion of these results in all cases to more carefully express appropriate caution (e.g., following the approach used in the current version of the abstract, which strikes a good balance).

We have included caveats for all discussions of the sustainability of macaque shellfish harvesting, to match the tone of the paper to that expressed in the Abstract. We have removed unfounded speculation on the unsustainability of shellfish predation, to better focus on the actual data produced by our study. We removed the quoted sentence from the Materials and methods section, and have integrated the same point within the Discussion, where it supports our argument that prey replenishment is similar between the two islands.

3) The estimate of how long before macaques could consume all periwinkles on the two islands, needs revision/clarity. If periwinkle snails are able to reproduce at the second year, this has to be accounted for in the prediction of the depletion, particularly for NomSao island.

We have clarified that our estimates of the periwinkle consumption rate were not intended to estimate the time until complete prey depletion on the islands, but instead they are hypothetical comparisons of annual consumption versus current estimated prey populations in a closed environment. The estimated rates are a guide to help readers understand the difference in predation intensity between the two islands, and its potential effects on prey depletion. We have also clarified that our estimates intentionally did not include population replenishment via planktonic larvae, as this replenishment occurs from ocean currents and are therefore similar between islands.